# Alternative Splicing for *Leucanthemella linearis NST1* Contributes to Variable Abiotic Stress Resistance in Transgenic Tobacco

**DOI:** 10.3390/genes14081549

**Published:** 2023-07-28

**Authors:** Hai Wang, Yuning Guo, Xueying Hao, Wenxin Zhang, Yanxia Xu, Wenting He, Yanxi Li, Shiyi Cai, Xiang Zhao, Xuebin Song

**Affiliations:** College of Landscape Architecture and Forestry, Qingdao Agricultural University, Qingdao 266109, China; wh672831@126.com (H.W.); 20212110007@stu.qau.edu.cn (Y.G.); 20212210002@stu.qau.edu.cn (X.H.); 17806260856@163.com (W.Z.); 20212110027@stu.qau.edu.cn (Y.X.); qauylhwt@163.com (W.H.); lyx33091@126.com (Y.L.); qaucaishiyi@163.com (S.C.); crazyz18@163.com (X.Z.)

**Keywords:** alternative splicing, *NST1*, abiotic stress, *Leucanthemella linearis*

## Abstract

*Leucanthemella linearis* is a marsh plant in the family Compositae. It has good water and moisture resistance and ornamental properties, which makes it one of the important materials for chrysanthemum breeding and genetic improvement. The *NST1* (*NAC* secondary wall enhancement factor 1) gene is associated with the thickening of the secondary walls of fiber cells in the plant ducts and the secondary xylem and plays an important role in plant stress resistance. In this study, two variable spliceosomes of the *NST1* gene were identified from a chrysanthemum plant by using bioinformatics, qRT-PCR, transgene, and paraffin section methods to explore the molecular mechanism of the variable splicing of *NST1* under abiotic stress. The results show that only three amino acids were found to be different between the two *LlNST1* variants. After being treated with salt, drought, and low temperatures, analysis of the expression levels of the *LlNST1* and *LlNST1.1* genes in *Ll* showed that *LlNST1.1* could respond to low temperatures and salt stress and had a weak response to drought stress. However, the expression level of *LlNST1* under the three treatments was lower than that of *LlNST1.1*. *LlNST1* transgenic tobacco showed increased saline–alkali resistance and low-temperature resistance at the seedling stage. *LlNST1.1* transgenic tobacco also showed enhanced saline–alkali resistance and drought resistance at the seedling stage. In conclusion, the functions of the two variable spliceosomes of the *NST1* gene are very different under abiotic stress. Therefore, this study verified the function of the variable spliceosome of *NST1* and improved the stress resistance of the chrysanthemum plant under examination by regulating the expression of the *NST* protein, which lays a material foundation for the improvement of plant stress resistance materials and has important significance for the study of the resistance of chrysanthemum plants to abiotic stress.

## 1. Introduction

*Leucanthemella linearis*, a perennial bog plant belonging to the Chrysanthemum subfamily of the Compositae family, has good waterlogging tolerance and ornamental characteristics, and it is an important resource for breeding chrysanthemum with waterlogging tolerance [1]. *Dendranthema morifolium*, a perennial herb of Chrysanthemum, is native to China [2]. With a long history of cultivation and rich germplasm resources, it is an important ornamental flower and plays a dominant role in the market [3,4]. With harsh changes in the environment and climate, it is necessary for plants’ environmental resistance to increase. Drought, low temperature, salinity, and other abiotic stresses are the main factors that limit the production and cultivation of plants, affecting the growth and development of chrysanthemums and leading to a decrease in yield [5]. Therefore, it is necessary to improve the resistance of *Ll* to enhance its environmental adaptability. When plants are subjected to abiotic stress, in order to maintain normal growth and development, the plant body will produce corresponding stress response mechanisms to reduce or eliminate the impact of these stresses [6]. 

Alternative splicing (AS) is an important mechanism that regulates gene expression and provides transcriptome and proteome diversity in plants under environmental stress [7]. It was found that approximately 61% of introns in normally growing *Arabidopsis thaliana* experienced variable splicing events, while AS events occurred more frequently under an abiotic stress [8]. For example, the adaptation of organisms to the environment enables precursor mRNA splicing to respond to abiotic stresses, including extreme temperature and drought [9]. Therefore, AS is closely related to plant growth and development processes and environmental adaptability [10]. Alternative splicing plays an important role in various reactions, such as photosynthesis, circadian rhythm, and stress response in plant growth and development stages [11,12,13], especially for some plants under abiotic stress. Studies on plants show that AS is involved in many plant responses under biological and abiotic stress, including high-temperature stress, cold stress, and drought stress [14]. At present, studies on the enhancement of plant stress resistance by alternative splicing mostly focus on Maize, Oryza sativa, tomato, and *At* but not *Compositae* [15,16,17,18].

Recent studies have shown that *MYB* (v-myb avian myeloblastosis viral oncogene homolog), *bHLH* (basic helix–loop–helix), *bZIP* (basic region-leucine zipper), *NAC* [NAM (no apical meristem), *ATAF* (Arabidopsis transcription activation factor), *CUC* (cup-shaped cotyledon)], and other transcription factors play a very important role in plant growth and development, pathogen threat, and stresses, such as low temperature and drought [19,20]. The *NAC* family is found only in plants and has a highly conserved N-terminal DNA-binding domain and a variable C-terminal regulatory domain. The *NAC* transcription factor plays an important role in regulating plant growth and development in response to abiotic stress [21]. The xylem development genetic network regulated by the *NAC* transcription factor gene has been characterized in *Populus triclocarbans* [20]. The *NST* (*NAC* secondary wall thickening promoting factor) transcription factor plays an important role in plant growth and development under abiotic stress. It was found that *NST1* and *NST2* regulate the secondary cell wall (SCW) thickening of the anther chamber wall [22]. *NST3* (*NAC* secondary wall thickening promoting factor 3)/*SND1* (secondary-wall-related *NAC* structure domain protein 1), *NST1*, and *NST2* are specifically expressed in fibrocytes and are involved in the thickening of fibrocyte secondary walls in the ducts and the secondary xylem [23,24,25]. This suggests that *NST* plays an important role in the regulation of lignin and cellulose synthesis. Overexpression of *ZmNST3/4* in maize can activate the expression of a series of synthetic genes related to cellulose and lignin, thus regulating the synthesis of cellulose and lignin in the maize secondary wall [26,27].

These results indicate that *NST* plays an important role in regulating lignin and cellulose synthesis by activating the expression of a series of related genes in the lignin and cellulose synthesis pathways of the plant's secondary wall. *Ll* is an important ornamental flower in Asteraceae. Lignin is an important component of the secondary wall of plant cells, and its content and the expression of related enzymes are important factors affecting plant morphology, stem hardness, and stress resistance. However, little is known about the molecular mechanism of *NST* transcription factors regulating lignin and cellulose synthesis in *Ll* [28]. *NST* genes play an important regulatory role in the synthesis of SCW in many species, but few studies have been conducted on the *NST* genes of *Ll.* In an earlier study, two alternative spliced transcripts of *LlNST* were cloned by RT-PCR. One of them, named *LlNST1*, and the other, *LlNST1.1*, has an alternative 3′ splice site. *Cynara scolymus* is a plant in the Compositae family and is closely related to *Ll*. This gene is highly similar to *CsNST1*. 

It was hypothesized that the two alternative spliceosomes of *LlNST* could play an important role in lignin and cellulose synthesis by regulating the expression of lignin and cellulose synthesis genes, thereby enhancing the stress resistance of *Ll*. Then, we can verify the function of the two alternative spliceosomes through the overexpression form, observe the phenotype of transgenic tobacco overexpressing lines under abiotic stress, and prove that the two alternative spliceosomes can enhance or weaken the stress resistance of transgenic tobacco. At the same time, wild-type *Ll* was treated with abiotic stress, and qPCR was used to test the expression changes of two alternative splices in *Ll* during abiotic stress. Then, we inferred whether the two alternative spliceosomes are involved in the process of plant resistance to abiotic stress and what their functions are.

In this study, real-time quantitative PCR was used to detect the expression of *NST* alternative spliceosomes in wild-type *Ll* under stress and the transgenic tobacco (*Nicotiana tabacum cv*. Nc89) under stress conditions to validate the functions of different alternative splicing of NST1.

## 2. Materials and Methods

### 2.1. Tobacco Transformation

In this study, the tobacco variety (Nicotiana tabacum cv. Nc89) and wild-type *Ll* were used as materials. The tissue culture seedlings of *Ll* used in this study were derived from the research group of Professor Zhao Huien, Beijing Forestry University. Total RNA was extracted using a FastPure^®^ Universal Plant Total RNA lsolation Kit (Vazyme, Nanjing, China) from *Ll*. Reverse transcription was performed using a HiScript III 1st Strand cDNA Synthesis Kit (+gDNA Wiper) (Nanjing Vazyme, China) and was eventually preserved at −20 °C. Based on the *CsNST1*(accession number LEKV01004794, NCBI) CDS sequences downloaded from the NCBI, the primers NST1-F (5′-ATGCTGCCCTCTCCTTTGAAT-3′) and NST1-R (5′-GCGAATTTGACCGGATTGG-3′) were designed to amplify *LlNST1* using cDNA from *Ll* as a template. The PCR fragments were obtained and cloned into the plant expression vector Super35S::GFP (Appendix A) by the double digestion technique. Plant expression plasmids were transferred into competent cells of the Agrobacterium tumefaciens strain GV3101. The transformed A. tumefaciens colonies were selected on LB-agar plates containing 50 mg/L of kanamycin, 50 mg/L of rifampicin, and 50 mg/L of gentamicin. The positive colonies were identified by PCR amplification of the inserted genes and used for the tobacco transformation as previously described.

### 2.2. Plant Material

Super35S::GFP(CK), Super35S::LlNST1, and Super35S::LlNST1.1 tobacco seeds were uniformly inoculated in screening medium (MS + 50 mg/LHgy + 30 g/L sucrose + 6 g/L AGAR) at ±25 °C and 60% relative humidity. The plants were cultured under a light intensity of 2500lx with a photoperiod of 16/8 h and treated when they produced 4–5 true leaves. The tobacco seedlings were moved to the treatment medium and the intelligent greenhouse of the College of Landscape and Forestry, Qingdao Agricultural University. Before stress treatment, the tobacco was grown to maturity using charcoal as a substrate [29].

### 2.3. Stress Management and Phenotypic Observation 

The tobacco transgenic plants at the seedling stage and the adult stage were treated with salt, drought, and low temperatures. 

#### 2.3.1. Abiotic Stress in Transgenic Tobacco at the Seedling Stage

The transgenic tobacco seedlings were treated with abiotic stress at the seedling stage, and the tobacco seedlings with the same growth rate were selected and transferred to the treatment medium, and the MS medium was selected as the basic medium. Approximately 200 nmol L^−1^ was used for salt treatment, 200 μmol L^−1^ ABA was used for drought treatment to simulate drought, and the basal medium was used for the low-temperature treatment and control. After 24 days of treatment, the plants were removed from the treatment medium, the agar carried by the roots was washed, and the surface water was blotted with filter paper. The fresh weight of the plants and the root length data were statistically measured.

#### 2.3.2. Abiotic Stress in Mature Transgenic Tobacco

At the same time, stress treatments were applied to the tobacco at the seedling stage. Five plants with similar growth were selected for each treatment of the Super35S::GFP, Super35S::LlNST1, and Super35S::LlNST1.1 lines. Salt treatment was simulated by applying 200 nm. L-1NaCl solution. The soil water content was less than 30% of the field water carrying capacity by using the limited watering method to carry out the special drought treatment. The samples were transferred to the low-temperature treatment using a low-temperature incubator at ±4 °C and light for 12 h. For the three treatments, the culture temperature ±25 °C and normal water supply were used as controls (CK).

After 15 days, the transgenic tobacco plants were photographed, statistical data were collected, and the tobacco plants treated with salt, drought, and low temperatures were moved to a greenhouse for rewatering. Seven days later, following rehydration, the tobacco transgenic plants were photographed, and related data were collected.

### 2.4. Paraffin Section

One-centimeter stems of the transgenic tobacco plants treated with abiotic stress were taken above the demarcation line where the tobacco stem was 1/3 of a distance from the soil. The sample was treated with FAA (70% alcohol: formalin: acetic acid 18:1:1) for 24 h and then soaked in a mixture of hydrogen peroxide and glacial acetic acid (1:1) to soften the material for 48 h, after which the sample was dehydrated with ethanol and embedded in paraffin. The microtome divided the sample into 10 µm slices. Finally, the sample was stained with saffron solid green. Microscopic examination, image acquisition, and analysis were carried out. Three biological replicates were set up for the experiment. The cell wall thickness of 10 cells in the visual field was measured randomly and analyzed statistically.

### 2.5. Stress Treatment

Sterile wild-type *Ll* plants were treated with salt, drought, and low temperatures. MS medium was selected as the base medium, 200 nm. L-1NaCl was used for the salt treatment, 15% PEG6000 was used for the drought treatment [30], and the base medium was used for the low-temperature treatment and the control. The whole plant was selected as the test sample. Samples were taken at seven stages of stress treatment time: 0 h, 1 h, 4 h, 8 h, 12 h, 24 h, and 36 h. The collected plant materials were frozen in liquid nitrogen before being stored in a −80 °C ultra-low-temperature refrigerator.

### 2.6. Sequence Alignment

Based on the amino acid sequences of *LlNST1* and *LlNST1.1*, the sequences were analyzed in detail through multiple alignments of nucleotide (amino acid) sequences using MEGA11.0.13. The sequences were compared using DNAMAN version 8.0 (Lynnon Biosoft, Quebec City, QC, Canada). cDNA sequencing revealed that *LlNST1* contained coding sequences that were not present in *LlNST1.1*.

### 2.7. qRT-PCR

Total RNA was extracted using a FastPure^®^ Universal Plant Total RNA lsolation Kit (Nanjing Vazyme, China) from *Ll*. Reverse transcription was performed using HiScript^®^ III RT SuperMix for qPCR (+gDNA Wiper) (Vazyme, Nanjing, China) and was eventually preserved at −20 °C. Quantitative primers for LlNST1 and LlNST1.1 were designed with Primer Premier version 5.0 (PREMIER Biosoft International, San Francisco, CA, USA). The whole cDNA was selected as the template for the qRT-PCR analysis of related genes. 2×ChamQ Universal SYBR qPCR Master Mix (Vazyme, China) was used. At ACTIN was used as the control gene. Gene expression was calculated using the 2^−∆∆CT^ method. There were three biological replicates and three technical replicates for each reaction. The qRT-PCR primers are listed in Appendix A.

### 2.8. Data Analysis 

All treatments mentioned in this study involved at least three independent biological and technical replicates. Microsoft Excel2016, GraphPad8.0.2, and IBM SPSS 27 were used for data statistics and analysis.

## 3. Results

### 3.1. Bioinformatics Analysis of the Variable Spliceosome of NST

Specific primers were designed to clone the *LlNST* cDNA, and two fragments of different sizes were amplified and sequenced. One open reading frame (ORF), named *LlNST1*, had 1113 bp and encoded 367 amino acids. The other, named *LlNST1.1*, had an ORF length of 1101 bp and encoded 364 amino acids. The results are shown in Figure 1a. The BLAST search of the NCBI protein sequence database showed that the amino acid sequence of *NST1* and *NST1.1* shared 82.39% amino acids, similar to *Cynara cardunculus var. scolymus* (L.) Fiori *NST1*. Figure 1b shows a schematic diagram of alternative splicing. Alternative splicing of *NST1* is mediated by alternative 3′ splicing, which is the most common type of alternative splicing in plants [31].

### 3.2. Phenotype of the NST Alternative Spliceosome under Stress Treatment in Tobacco

The transgenic tobacco plants overexpressing *LlNST1* and *LlNST1.1* acquired higher stress tolerance by increasing their root length and fresh weight [18]. To investigate the functions of *NST1* and *NST1.1* under abiotic stress, the transgenic tobacco plants were subjected to salt, drought, and low-temperature stress, and the growth differences of the transgenic plants under different treatments were observed (Figure 2). 35S::GFP was set as the control group, and *LlNST1* and *LlNST1.1* were used as the experimental groups. The growth data of the tobacco at the seedling stage were counted after 24 days of stress treatment, and the fresh weight and root length of the individual plants were counted (Figure 3).

As shown in Figure 2 and Figure 3, the three transgenic tobacco seedlings subjected to the three stress treatments were compared 24 days after the stress treatments were applied to the tobacco plants. Under the control treatment, compared with Super35S::GFP transgenic tobacco, the weight of *LlNST1* decreased by 10%, and that of *LlNST1.1* decreased by 55%, and the difference reached a significant level, while the root length did not change. Under the simulated drought treatment with 200 μmol/L ABA in the basal medium, the individual weight of the three transgenic tobacco treatments did not change significantly. It is worth noting that the root length of Super35S::GFP transgenic tobacco increased by 43% compared with *LlNST1.1*, and the difference reached a significant level. Under salt stress, compared with the Super35S::GFP control group, the individual weight of *LlNST1* and *LlNST1.1* increased by 238% and 207%, respectively. The *LlNST1.1* transgenic tobacco had a 29% shorter root length than Super35S::GFP, and the difference reached a significant level. Under low-temperature stress, compared with Super35S::GFP, the individual weight and root length of tobacco in *LlNST1* and *LlNST1.1* decreased, and the individual weight of *LlNST1.1* decreased by 72% compared with Super35S::GFP. Compared with Super35S::GFP, *LlNST1* and *LlNST1.1* had 36% and 74% shortened root lengths, respectively, which reached a highly significant level. 

The experimental results show that under stress treatment, the growth rate of *LlNST1* and *LlNST1.1* decreased compared with the control. The ability of *LlNST1* to resist saline–alkali stress was enhanced, and that of *LlNST1.1* to resist drought was also enhanced.

To observe the phenotypic changes of alternative spliceosomes under stress more intuitively, the growth patterns of the transgenic tobacco under stress treatment were observed (Figure 4), and the stem height changes in the tobacco during stress and after rewatering after stress were examined (Figure 5). 

In Figure 5, the phenotypic trait analysis under different forms of stress and the height changes of the transgenic tobacco plants observed under the different stress treatments are shown. Under the control treatment, the average plant height of the *LlNST1* transgenic tobacco was 2.08 cm shorter than that of the control. After rewatering treatment, the average plant height of the *LlNST1* transgenic tobacco was 1.04 cm shorter than that of the control, and the growth rate of the transgenic tobacco was significantly weakened. The average plant height of the *LlNST1.1* transgenic tobacco was 1.34 cm shorter than that of the control, and after rehydration, the average plant height was 0.84 cm shorter than that of the control, and the growth rate of the transgenic tobacco was significantly weakened. Under salt treatment, the *LlNST1* transgenic tobacco plants were 0.64 cm shorter than the control tobacco plants. The average plant height of the *LlNST1* transgenic tobacco increased by 0.42 cm compared with the control tobacco after rehydration. The salinity tolerance of the *LlNST1* transgenic tobacco was significantly reduced. The *LlNST1.1* transgenic tobacco plants increased by 0.88 cm and 1.68 cm in height on average compared with the control tobacco plants. The salinity tolerance of the *LlNST1.1* transgenic tobacco was significantly enhanced. Under low-temperature treatment, the height of the *LlNST1* transgenic tobacco increased by 0.9 cm and 1.04 cm compared with the control. The *LlNST1* transgenic tobacco showed significantly increased tolerance to low temperatures. The average plant height of the *LlNST1.1* transgenic tobacco was 0.08 cm longer than that of the control group. After rewatering treatment, the plant height of the *LlNST1.1* transgenic tobacco was 0.18 cm shorter than that of the control group, and the low-temperature tolerance of the *LlNST1.1* transgenic tobacco was significantly weakened. Compared with the control group, *LlNST1* and *LlNST1.1* reduced the growth rate and drought resistance of the tobacco, but *LlNST1* enhanced the resistance to low temperatures, and *LlNST1.1* enhanced the resistance to salt and alkali conditions. *LlNST1* and *LlNST1.1* may enhance the stress resistance of *Ll*. 

### 3.3. Stem Anatomy Analysis of Transgenic Tobacco

The epidermis, cortex, phloem, xylem, and pith could be clearly seen in the cross-section at one-third of the stem of the 12-week-old transgenic tobacco. Among them, the part stained red by safranin fast green is the lignified cell wall and tube, and the part-stained green is the plant cellulose cell wall, and the sieve tube is shown in Figure 6. The cell wall thickness of these three parts of the tissue was measured and compared in the 35S::GFP(CK), *LlNST1*, and *LlNST1.1* transgenic tobacco, and the statistical analysis is shown in Figure 7. The results showed that there were significant differences in xylem cell wall thickness among the three transgenic tobacco species (*p* < 0.05). CK was the thickest, followed by *LlNST1* and *LlNST1.1*. There was no significant difference in the size of the xylem cells.

### 3.4. Analysis of qRT-PCR Results of Leucanthemella Linearis under Different Treatments

To further explore the impact of the *LlNST1* and *LlNST1.1* genes on the resistance of *Ll*, a comparative analysis of the expression levels of the *LlNST1* and *LlNST1.1* genes in the wild-type *Ll* under stress in the different treatments was conducted (Figure 8).

The expression levels of *LlNST1* and *LlNST1.1* did not change significantly under 15% PEG6000 simulated drought treatment. At 4 °C, *LlNST1* decreased rapidly after the time period of 8 h was raised to 12 h, and the total volume of *LlNST1.1* was higher than *LlNST1.* In the case of salt treatment, the amount of *LlNST1* did not significantly change, and its expression was low, the level of expression at 4 h was sharply reduced, and it was 7.5 times the level of the 0 h expression. The *LlNST1* and *LlNST1.1* levels of gene expression in the three kinds of stress treatment decreased, but the expression of *LlNST1.1* was higher than *LlNST1*. 

The analysis shows that *LlNST1.1* can respond to salt and low-temperature stress and that it shows a weak response to drought stress.

## 4. Discussion

The growth, development, reproduction, and production of plants are severely affected by the comprehensive harmful effects of high salt and drought conditions [32]. Only gaining a better understanding of the tolerance mechanism of plant environmental stress can help us to find ways and strategies to improve plants’ resistance to abiotic stress [33]. Therefore, this study reports the expression of the *LlNST1* variable splicing body and improves the resistance of *Ll*. The *NST1* gene and family analysis found that the *LlNST1* gene was more similar to the *Cs* and the experimental results showed that *LlNST1* and *LlNST1.1* were associated with resistance to abiotic stress and that *LlNST1* was functional in terms of the results reported by *NST*; thus, the two *NST1* variants were named *LlNST1* and *LlNST1.1*. We identified two *LlNST1* variants in *LlNST1* gene cloning and proposed the possibility of the selective splicing of *LlNST1*.

We tested the function of the two-alternative spliceosomes by examining the phenotypes of transgenic tobacco overexpressing lines under abiotic stress. The change in the phenotype during treatment of the tobacco phenotype shows that the growth rate of the *LlNST1* splicing body *LlNST1.1* is reduced and the resistance to drought is weakened, and the resistance to low temperatures in *LlNST1* increases the ability of *LlNST1.1* to strengthen the resistance to salt. These results demonstrate that the two alternative spliceosomes can enhance the stress resistance of transgenic tobacco. At the same time, the expression of *LlNST1* spliceosomes under abiotic stress was identified and analyzed in the wild-type *Ll*. qPCR was used to examine the expression changes of the two-alternative splicing *Ll* during the process of abiotic stress. The results of real-time fluorescence quantitative PCR showed that *LlNST1.1* could respond to salt and low-temperature stresses, while *LlNST1* expression was too low to be used as a real and effective experimental basis. To further verify the function of the two-alternative spliceosomes, the secondary cell wall and lignin synthesis of overexpressed tobacco were observed using the paraffin section technique. Studies have shown that the overexpression of *NST* in plants, such as *At*, *Zm*, and *Pt,* results in the thickening of the secondary wall of fiber cells and the content of cellulose and lignin in the cell wall is significantly increased [25,26,34]. However, the results of the stem dissection experiment showed that the xylem cell wall thickness in the transgenic tobacco experiment with overexpression of the two-alternative spliceosomes of the *NST1* gene was not significantly increased but rather decreased. It is speculated that *NST1* variants may play a negative role in the lignin regulatory network or have identical or redundant genes in the regulatory network.

In a study on the transcription control network that is reported to regulate SCW biosynthesis, *NST1* and *SND1* are the main transcriptional switches of the development program that regulate SCW biosynthesis by activating a series of downstream transcription factors [35,36]. In *Cucurbita moschata*, the key role of *CmNSTl* as a master regulator of the lignin biosynthesis pathway during seed coat development was revealed, while other NAC and MYB transcription factors are also involved in forming regulatory networks for secondary cell wall construction [37]. The study found that both snd1 and *NST*1/2 were found to have a severe defect in the plant and in the phloem of the cotton stem, but the individual silence *SND1*s or *NST1*s did not have a distinct phenotype. This means that *SND1*s and *NST1*s are involved in SCW growth [38]. Monogenetic mutants usually show no phenotype [39]. This shows that the regulatory network is flexible and complex. The results of the experiment showed that the anti-inversion was enhanced. 

Interestingly, in the phenotype observation of transgenic tobacco, we found that the growth rate of the transgenic tobacco was reduced, but the growth potential was not reduced, and the stress resistance was enhanced. We speculate that this phenomenon is caused by the weakening of the plant growth rate in order to resist the external environment, which needs further exploration in the future.

Because the genetic transformation of tobacco is relatively mature and the experimental period is short, the genetic transformation system of *Ll* has not been established, so the gene was not transferred into this species for overexpression experiments. Future experimental progress will continue to improve the genetic transformation system, the molecular mechanism, and the downstream network of NST1 variable splicing in *Ll.*

## 5. Conclusions

In this study, two alternative spliceosomes of *LlNST1* were cloned, and the splicing type was identified as A3SS. Overexpression of *LlNST1* and *LlNST1.1* in transgenic tobacco plants can enhance the stress tolerance of transgenic tobacco, and *LlNST1* is resistant to low temperatures, and *LlNST1.1* is resistant to salt and alkali conditions. The transection structure of the transgenic tobacco stems was observed by the paraffin section technique to observe the effect of overexpression on lignin and cellulose synthesis. The thickness of the xylem cells was related to the growth rate of the plant, with thick cell walls due to fast growth rates and thin cell walls due to slow growth rates. However, the relationship between lignin and cellulose synthesis and abiotic stress remains unclear. Future experiments will continue to explore the molecular mechanism between the two and obtain a complete regulatory network. In the future, we will establish the genetic transformation system of *Ll* to promote its application so that more resistant water-tolerant chrysanthemum cultivars can bloom everywhere.

## Figures and Tables

**Figure 1 genes-14-01549-f001:**
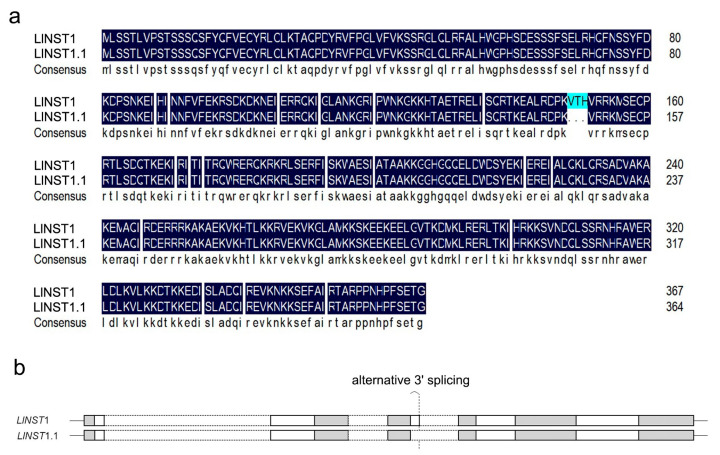
Alignment of the amino acid sequences of the two *LlNST* genes with a schematic representation of alternative splicing. (**a**) Sequence alignments of the proteins are shown in the figure. Identical amino acids are shaded in black and conserved changes are shaded in blue. (**b**) *LlNST1* and *LlNST1.1* alternative splicing diagram; the gray parts represent exons, and the white dashed boxes represent introns. The white part represents the mode of alternative splicing selection.

**Figure 2 genes-14-01549-f002:**
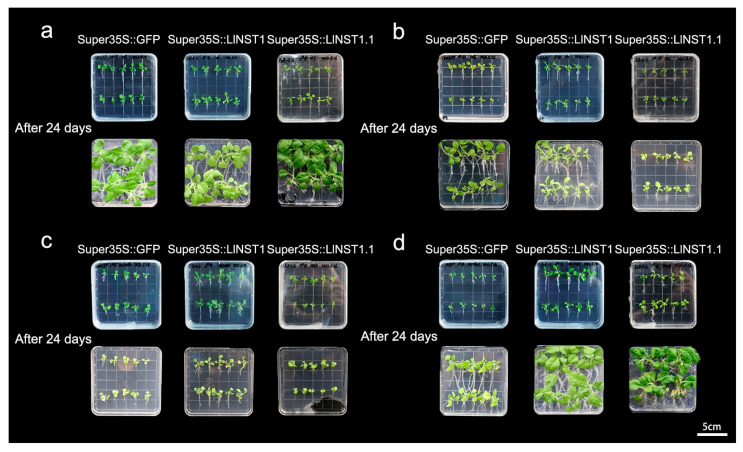
Growth state of the tobacco transgenic plants. (**a**) For the control treatment, MS medium was used and placed at 25 °C; (**b**) for the low-temperature treatment, MS medium was used and placed at 4 °C; (**c**) for the drought treatment, 200 μmol/L ABA was added to the MS medium at 25 °C; and (**d**) for the salt treatment, MS medium was used with 200 nmol/L NaCl added and placed at 25 °C. Scale bars, 5 cm.

**Figure 3 genes-14-01549-f003:**
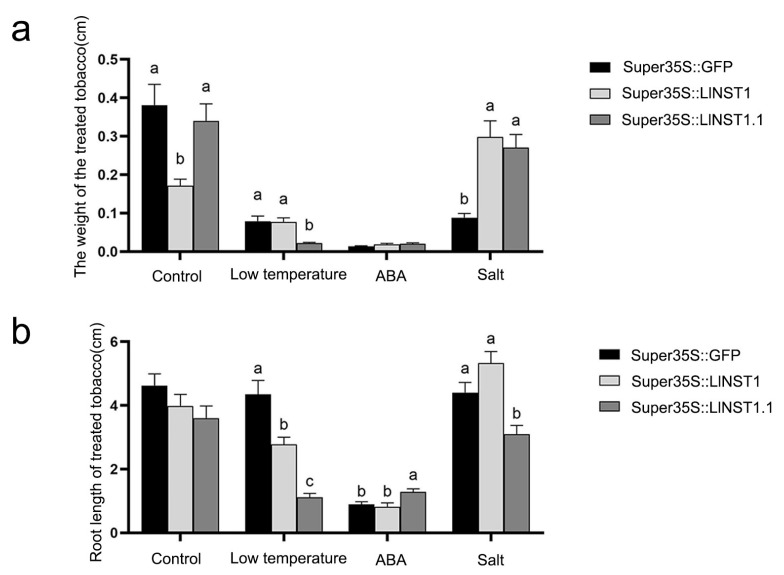
Fresh weight and root length of the transgenic tobacco after stress treatment. (**a**) Fresh weight per plant of transgenic tobacco at the seedling stage after four treatments and (**b**) Root length per plant of three transgenic tobacco plants after four treatments. Note: The different letters indicate the significance of the difference (*p* < 0.05).

**Figure 4 genes-14-01549-f004:**
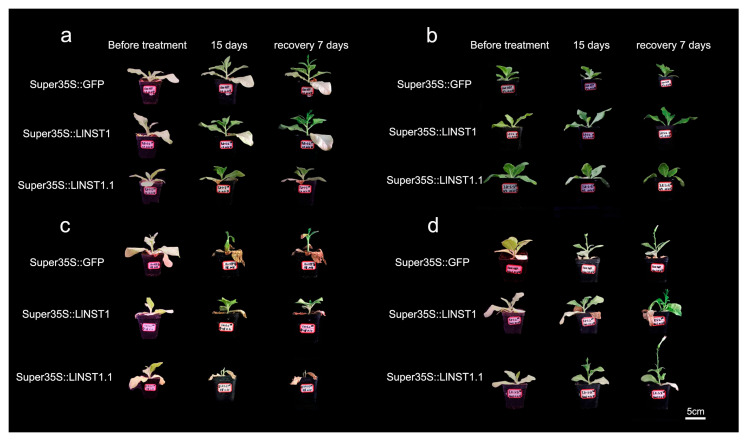
The phenotype of transgenic tobacco after stress treatment and rehydration. Denoted are the plant phenotypes after 15 days of treatment versus the phenotypes after 7 days of rehydration. (**a**) The control treatment at 25 °C; (**b**) the low-temperature treatment at 4 °C; (**c**) the drought treatment at 25 °C; and (**d**) the salt treatment using 200 mmol/L NaCl at 25 °C. Scale bars, 5 cm.

**Figure 5 genes-14-01549-f005:**
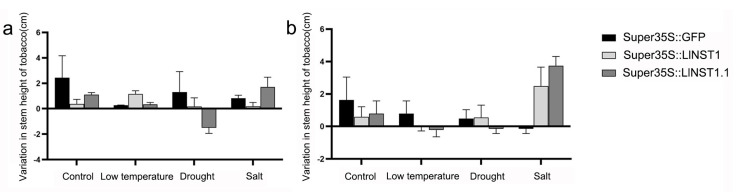
Stem height changes in transgenic tobacco after stress, during stress, and after rehydration. (**a**) shows the change in the stem height of the tobacco after stress treatment, and (**b**) shows the change in the stem height of the tobacco after 7 days of rehydration following stress treatment.

**Figure 6 genes-14-01549-f006:**
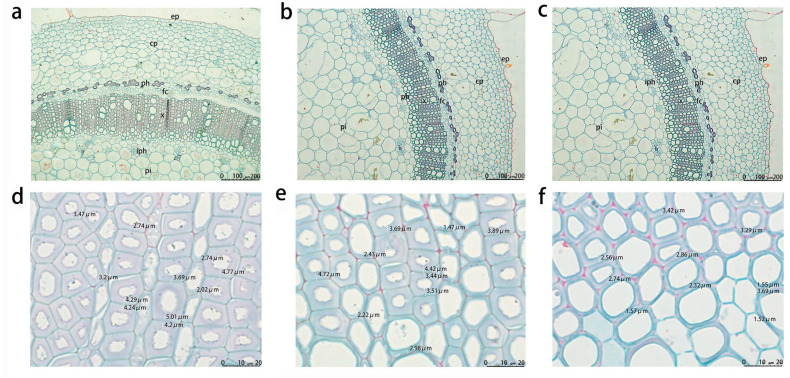
Micrographs showing the stem anatomy of the 35S::GFP, *LlNST1*, and *LlNST1.1* transgenic tobacco in the normal growth state. (**a**,**b**) show the anatomical structure of the 35S::GFP transgenic tobacco stems; (**c**,**d**) show the anatomical structure of the stem xylem of the *LlNST1* transgenic tobacco; and (**e**,**f**) show the stem anatomy of the *LlNST1.1* transgenic tobacco. Detailed are the tissue locations shown with safranin fast green staining. ep = epidermis, cp = cortical parenchyma, ph = phloem, iph = internal phloem, fc = cambium, x = xylem, and pi = pith.

**Figure 7 genes-14-01549-f007:**
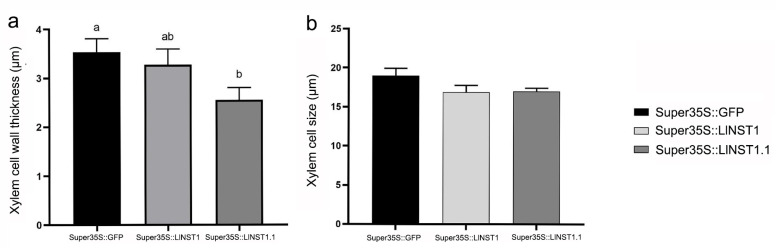
Xylem cell wall thickness and xylem cell size in tobacco of different genotypes. (**a**) Xylem cell wall thickness in tobacco of different genotypes. (**b**) Xylem cell size in tobacco of different genotypes. Note: The different letters indicate the significance of the difference (*p* < 0.05).

**Figure 8 genes-14-01549-f008:**
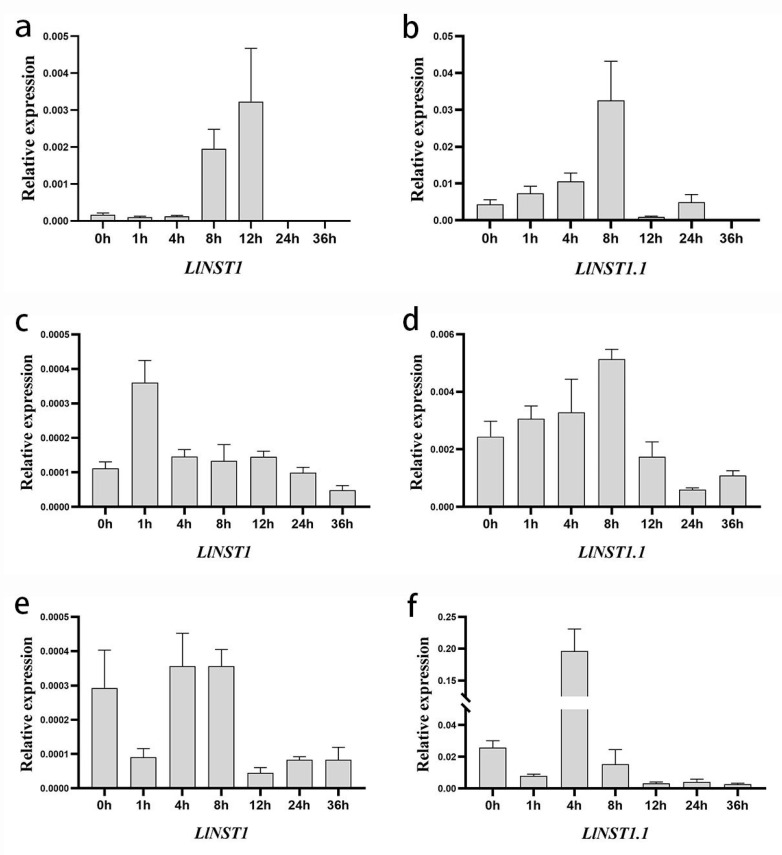
The expression levels of LlNST1 and LlNST1.1 in *Ll* were analyzed under low-temperature, drought, and salt stress conditions. (**a**) Expression of *LlNST1* in *Ll* under low-temperature stress. (**b**) Expression of *LlNST1.1* in *Ll* under low-temperature stress. (**c**) Expression of *LlNST1* in *Ll* under drought stress. (**d**) Expression of *LlNST1.1* in *Ll* under drought stress. (**e**) Expression of *LlNST1* in *Ll* under salt stress. (**f**) Expression of *LlNST1.1* in *Ll* under salt stress.

## Data Availability

Data is contained within the article or Appendix A. The data presented in this study are available in [insert article or Appendix A here].

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
