# Peer review of "Alternative Splicing for Leucanthemella linearis NST1 Contributes to Variable Abiotic Stress Resistance in Transgenic Tobacco"

_genes, 2023, doi:10.3390/genes14081549_

Round 1

Reviewer 1 Report

The manuscript by Wang and collaborators presents an experimental analysis pertaining the effect of two splice variants isolated from a wild relative of chrysanthemum (L. linearis) and their role in the response to different kinds of abiotic stress. As such, the subject and approach are interesting, nevertheless, in it current form the manucript is lacking a coherent outline and presentation, needing extensive editing for clarity in all sections, in particular the methods, results and discussion sections. For instance, the Methods section is lacking many important details that directly impact the results and the soundness of the conclusions drawned from them. Below I expand in some of the core issues of the manuscript but also, please refer to individual comments in the annotated pdf version provided, where in yellow are important statements and in blue background are words and/or phrases that are incorrect or are unclear.

General observations:

Abstract:

Needs re-working to make more explicit if differences in stress response between LiNST1 and LiNST1.1. are consistent across the two experimental systems used (chrysanthemum; L. linearis and tobbaco) or not. Currently, the abstract reads as if differences were only found in the chrysanthemum background.

Introduction:

Needs to be reorganized, starting with the subject species from where the splice variants were isolated (L. linearis), and then talk about chrysanthemums. Also, review phrases marked in blue, as they do not add to the introduction and seem infinished or have inconsistencies in articles and plurals.

Revise and put in italics the scientific names for Arabidopsis and Populus.

Put complete name of genes at first mention.

Methods section:

Methods are very poorly described throught this section and many important information of missing. Some of it is:

Why tobacco was used as an heterologous system? Furthermore, the method of transformation and the backbone of the construction used are not disclosed, as well as if the tobacco plants where transformed with the cDNAs of L. linearis, or gDNAs. Also, it is incomprehensible why no real wild type tobacco plants (non-gm) were used and instead, a reporter line (35S::GFP) was used as a “control”.  In this line of thought, analyses of the ortholog(s) of the target gene should have been performed in all lines, so as to show that the endogenous gene was not viasing results.

While several histological measurements were done, it is not clear why no qPCR analyses were performed in the stressed tobacco lines, such as those performed on the L. linearis plants from were the 2 splice variants were originally isolated.

Additionally, it is not clear why more mature tobacco plants where analyzed histologically (24+ days), while the experiment involving qPCR in L. linearis seems to have been performed in plantlets (as they are reported to have been grown in MS medium). In this same line of thought, it is not disclosed in what stage this plantlets were subjected to different abiotic stresses, but being in MS medium precluded a comparable simulation of drought, as carried out with the tobacco plants. Thus, results are not comparable. Furthermore, qPCRs have results that seem to be near the LOD of a qPCR machine and as such, are questionable.

Results:

The writing of the results section is very confusing and hard to follow.

Discussion: the conclusions drawn by the authors are not supported by the data presented.

The English in the manuscript needs some editing, nevertheless, the overarching issue is a lack of clarity throughout the different sections.

Author Response

Thank you for your letter and for reviewer’s comments concerning our article entitled - Leucanthemella linearis NST1 gene variable splicing in abiotic stress resistance. (original submission ID: genes-2440670). Those comments are all valuable and very helpful for revising and improving our paper.

We have studied comments carefully and have made correction which we hope meet with approval. Revised portion are marked in yellow in revised paper (Article revision:Response to Reviewer Comments #1). All changes are shown in yellow with a highlighted background. The main corrections in the article and the responds to the reviewer’s comments are as flowing.

Reviewer 2 Report

The aim of this study was to evaluate the response of Leucanthemella linearis NST1 to abiotic stresses (temperature, drought and salinity) after changing the mechanism that regulates gene expression (NST1). The morphological response of the plants varied depending on the type of stress.

In general, the work is written correctly. However, it contains deficiencies, which should be improved.

1. Lack of a research hypothesis. Should be supplemented.

2) I propose to add a chapter " Summary" as a synthesis of the conducted research.

3. The discussion of the results is not very accurate. Supported by too little literature. Please add the latest literature on this research problem.

Author Response

Thank you for your letter and for reviewer’s comments concerning our article entitled - Leucanthemella linearis NST1 gene variable splicing in abiotic stress resistance. (original submission ID: genes-2440670). Those comments are all valuable and very helpful for revising and improving our paper.

We have studied comments carefully and have made correction which we hope meet with approval. Revised portion are marked in yellow in revised paper (Article revision:Response to Reviewer Comments #2). All changes are shown in yellow with a highlighted background. The main corrections in the article and the responds to the reviewer’s comments are as flowing.

Reviewer 3 Report

The article Leucanthemella linearis NST1 gene variable splicing in abiotic stress resistance is well executed and properly elaborate. The authors enlighten the spliceosome of NST1 and improved stress tolerance. The article has innovation and novelty but some changes must be required:

English is hard to understand. The sentences are not making sense at some points.

The objective of the study must be more elaborated.

The materials and methods section is so confusing, I recommend explaining the stress treatments individually with separate sub-headings.

A detailed method for stem microscopy must be added.

The authors should add the lettering on all graphs, some figures don't have lettering. I would recommend adding p-value on each figure to better understand results.

The take-home message is missing.

Needs to be improved

Author Response

Thank you for your letter and for reviewer’s comments concerning our article entitled - Leucanthemella linearis NST1 gene variable splicing in abiotic stress resistance. (original submission ID: genes-2440670). Those comments are all valuable and very helpful for revising and improving our paper.

We have studied comments carefully and have made correction which we hope meet with approval. Revised portion are marked in yellow in revised paper (Article revision:Response to Reviewer Comments #3). All changes are shown in yellow with a highlighted background. The main corrections in the article and the responds to the reviewer’s comments are as flowing.
